# Metacognition as a Transdiagnostic Determinant of Recovery in Schizotypy and Schizophrenia Spectrum Disorders

**DOI:** 10.3390/bs14040336

**Published:** 2024-04-17

**Authors:** Courtney N. Wiesepape, Elizabeth A. Smith, Jaclyn D. Hillis-Mascia, Sarah E. Queller Soza, Madyson M. Morris, Alison V. James, Alexis Stokes

**Affiliations:** 1Austin VA Clinic, Veterans Affairs Central Texas Health Care, Austin, TX 78744, USA; 2Department of Psychology, Indiana State University, Terre Haute, IN 47809, USA; liz.smith@indstate.edu (E.A.S.); madyson.morris@va.gov (M.M.M.); astokes10@sycamores.indstate.edu (A.S.); 3Chillicothe VA Medical Center, Chillicothe, OH 45601, USA; jaclyn.hillis-mascia@va.gov; 4Richard L. Roudebush VA Medical Center, Indianapolis, IN 46202, USA; sarah.quellersoza@va.gov; 5VA Maryland Healthcare System, Baltimore, MD 21201, USA; alison.james@va.gov

**Keywords:** schizophrenia spectrum disorders, schizotypy, metacognition, recovery, early intervention, psychotherapy

## Abstract

The term schizotypy refers to the latent personality organization that is thought to underpin the liability to develop schizophrenia and associated disorders. Metacognition, or the ability to understand and form increasingly complex and integrated ideas of oneself, others, and one’s community, has been proposed to be an important transdiagnostic construct across schizophrenia spectrum disorders and a range of both clinical and non-clinical manifestations of schizotypy. In this paper, we review evidence that deficits in metacognition are present in individuals with relatively high levels of schizotypy and that these deficits are related to symptomology, function, and quality of life. We address the idea that decrements in metacognition may also contribute to the progression from schizotypy to more severe manifestations, while the amelioration of these deficits may enhance aspects of recovery, including the ability to form an integrated sense of self, others, and the wider world. We also review the following two recovery-oriented psychotherapies that target metacognition to promote recovery in individuals with clinical manifestations of schizotypy: Evolutionary Systems Therapy for Schizotypy (ESTS) and Metacognitive Reflection and Insight Therapy (MERIT).

## 1. Introduction

Historically, schizophrenia and related disorders were viewed as deteriorating illnesses from which recovery was nearly impossible. However, both quantitative and qualitative research support the idea that individuals diagnosed with schizophrenia spectrum disorders recover more often than not, achieving symptom remission and functional improvement [1,2,3,4]. Although it is now widely accepted that recovery from schizophrenia and related disorders is possible, some research suggests that the proportion of individuals diagnosed with schizophrenia who recover has not increased despite major changes in interventions and treatment options [5]. There is no agreement on why this may be; however, it may indicate a lack of consensus on what recovery is and how to measure it [6,7], with some suggesting that rates of recovery from schizophrenia spectrum disorders vary significantly depending on how recovery is defined and the target population [8,9].

To better understand and measure recovery, researchers have conceptualized recovery to include both objective (e.g., symptom severity and level of functioning) and subjective (e.g., quality of life beyond clinical factors) aspects of an individual’s life [10]. These two domains are often viewed as complementary but not synonymous [10,11]. Supported by research illustrating that recovery is possible, various recovery frameworks focusing on subjective aspects have emerged. The CHIME framework [12], for example, is a widely accepted model of recovery that suggests that subjective markers of recovery include connectedness, hope, identity, meaning, and empowerment. Leonhardt et al. [11] further suggest that the need for meaning-making lies at the intersection of subjective and objective approaches to recovery in serious mental illness, including schizophrenia spectrum disorders. Although the subject of meaning-making (e.g., symptoms vs. other psychosocial challenges) may differ between objective and subjective frameworks, recovery is essentially conceptualized as meaning-making in the face of various threats or difficulties (e.g., stigma, trauma).

Despite agreement that recovery from psychosis is possible, some argue that our current interventions and treatments have yet to improve outcomes [5]. One proposed method to improve recovery outcomes is through early intervention. For example, delayed intervention studies have demonstrated that individuals diagnosed with schizophrenia who receive antipsychotic treatment within six months have better outcomes than those treated 6–12 months after the onset of symptoms [13,14]. Li et al. [9] found that a shorter duration of untreated psychosis was predictive of better outcomes in a population of individuals experiencing their first episode of psychosis. Specialized treatment or coordinated specialty care (CSC)—an evidence-based, multidisciplinary team approach that includes medication and health support, psychotherapy, family education, peer support, supported education and employment, and case management—also seems to play a role in improving recovery rates in early intervention, and has been associated with higher rates of engagement and improvement in objective recovery measures [15]. For example, individuals diagnosed with early psychosis who received comprehensive care through the Recovery After an Initial Schizophrenia Episode (RAISE) Early Treatment Program experienced decreased hospitalization, improved quality of life, and increased involvement in work and school when compared to those who received community care [16]. Although intervention in ultra-high-risk populations has been controversial [17], a meta-analysis demonstrated risk reduction in the transition to psychosis at 12-month and 24–48-month time periods after ultra-high-risk intervention, including antipsychotics and psychological intervention [18].

As reviewed above, early intervention in schizophrenia spectrum disorders has been shown to result in improved outcomes and higher rates of recovery. However, the term “early intervention”, like “recovery”, lacks a clear definition and framework. For example, early intervention is often used to describe both efforts to prevent the transition to psychosis in high-risk populations as well as efforts to treat psychotic disorders quickly once they emerge. McGorry and colleagues [19] suggest using a clinical staging model in which early intervention is defined and delivered based on the stage of illness. In this model, stage one includes ultra-high-risk populations, stage two includes first-episode psychosis, and finally, stage three includes the first 5 years after diagnosis. Although treatment is often complicated in these groups, it is suggested that certain interventions (e.g., fatty acids, psychotherapy) should be used in the early stages of high-risk populations while other interventions (e.g., psychotropics) can be used in later stages or when the aforementioned treatments have failed [20]. Generally, interventions that are considered to be more innocuous are used before interventions that may have greater side effects or risk for long-term iatrogenic effects.

In order to intervene in the early stages of the development of schizophrenia and related disorders, there must be a way to identify those at risk of developing these disorders and those in the early stages of displaying clinically meaningful symptomology that does not yet rise to the level of psychotic experiences. The study of schizotypy has shown promise in improving the early identification of and intervention in schizophrenia-related disorders. In this paper, we propose that metacognition is a transdiagnostic determinant of recovery, as improved metacognitive functioning is connected to recovery in a range of psychological disorders, and that targeting impaired metacognition represents an understudied method of early intervention in schizotypy populations. We first review the construct of schizotypy, including definitions and how it differs from an ultra-high-risk population. Next, we introduce metacognition and review evidence that supports its use as a transdiagnostic construct that promotes recovery. Finally, we consider the intersection between schizotypy and metacognition and review two psychotherapies that can address metacognition in schizotypy.

## 2. Schizotypy

Schizotypy is defined as the latent personality organization that is thought to underpin the liability to develop schizophrenia spectrum disorders [21,22]. More broadly, the term schizotypy is used to refer to a set of enduring personality traits that reflect a continuum from social eccentricity (e.g., increased creativity, positive spirituality) [23] to forms of thought disorder and social detachment, often coupled with magical thinking or paranoia [24]. Recent research from both phenomenological [25] and genetic perspectives [26] supports this dimensional model of schizotypy.

Schizotypy shares a substantial degree of phenomenology with schizophrenia spectrum disorders [25], consistent with its broad conceptualization as an attenuated expression of the symptoms of schizophrenia [21]. As such, schizotypy can be conceptualized as part of a multidimensional continuum of signs and symptoms, which includes schizotypal [27], paranoid, schizoid, and avoidant personality disorders [28], prodromal symptoms, schizophrenia, and other psychotic illnesses. Of note, each “disorder” on this continuum is thought to share significant conceptual overlap in symptomology (e.g., positive, negative, and disorganized domains), etiology, and phenomenology. The mild end of the spectrum involves subclinical, psychotic-like signs and symptoms. Because schizotypy can manifest prior to or in the absence of impairing, clinically significant symptoms, non-clinical schizotypy must be detected by trained observers using validated psychometric measures or interviews of schizotypal traits [21]. Studies of the latent structure of schizotypal traits generally support three-factor symptom solutions, including positive (odd experiences or beliefs, magical thinking, perceptual disturbances), negative (lack of volition or expression of affect, social detachment), and disorganized (stereotyped or concrete thinking) domains [29,30].

Schizotypal and other related personality disorders represent clinical disorders that fall between subclinical presentations and psychotic disorders (e.g., schizophrenia) on this continuum. For example, schizotypal personality disorder is often linked to schizotypy and involves pronounced impairment in interpersonal functioning, a lack of close relationships, and intense discomfort and suspiciousness around others. In schizotypal personality disorder, interpersonal anxiety is largely due to suspiciousness and paranoid ideations. Perceptual anomalies, ideas of reference, and unusual beliefs tend to be present, as well as idiosyncratic speech, odd appearances or mannerisms, and reduced or inappropriate affective expression. Generally, personality disorders remain relatively stable over time, but a small portion of individuals with schizotypal or other related personality disorders proceed to develop schizophrenia or a related disorder.

The concept of schizotypy originated in early family studies of schizophrenia when attenuated forms of schizophrenia pathology were observed in biological relatives of individuals with the disorder. This line of research led to using validated psychometric measures or interviews of schizotypal traits [21] to identify schizotypes in the general population that are psychosis-prone and theoretically share at least some of the genetic basis for schizophrenia. These psychometrically identified psychosis-prone individuals may display manifestations in one of the three symptom domains, including unusual thinking, perceptual anomalies, or asocial tendencies, but may not have notable distress or functional impairment.

Each domain of schizotypy increasingly appears to have its own course and association with the development of schizophrenia spectrum disorders. Positive schizotypy has consistent associations with psychotic-like prodromal symptoms and traits of schizotypal personality disorder across multiple studies [31] but does not necessarily involve substantial functional impairment, perhaps because the symptoms may be experienced as benign (for example, in cases of unusual beliefs that are experienced as helpful) [31,32]. Disorganized schizotypy is associated with functional impairment, as well as depression and traits of borderline, avoidant, and paranoid personality [31]. Negative schizotypy is associated with deficits in functioning and schizoid personality traits, as well as schizotypal and paranoid traits and no history of dating relationships. Compared to positive and disorganized schizotypy, negative schizotypy is particularly difficult to capture in individuals in the general population, which could be a result of avoiding research participation due to social anxiety [31] or reduced awareness of a lack of affective expression or other deficit symptoms, which would especially affect detection with self-report questionnaires [33].

Studies combining the psychometric framework with family and genomic approaches have supported the association of schizotypy with schizophrenia through familial pathways. The positive dimension appears to be subject to an additive genetic effect in which higher levels of schizotypy are associated with a higher polygenic burden of genes involved in dopamine signaling [26]. Likewise, negative schizotypy demonstrates associations with genes implicated in schizophrenia. One study found that in non-psychotic family members, interview-assessed negative schizotypy was associated with polygenic risk scores for schizophrenia but not positive or disorganized schizotypy [33]. Novel frameworks indexing environmental risks, in addition to genetic risks, suggest that the expression of schizotypy depends on the interaction of both genetic and environmental factors. Specifically, one study found that genetic loading moderates the effects of environmental stress on the expression of schizotypal traits across positive, negative, and overall schizotypy in individuals unaffected by illness [34].

Relatedly, early intervention studies of ultra-high-risk populations utilize a similar attenuated-symptom set of criteria to explicitly identify individuals in the prodrome of schizophrenia, as noted previously [19]. Although there is considerable overlap between schizotypy and ultra-high-risk populations, there are also important differences. To be considered ultra-high risk, individuals must be of a certain age and meet one of three specific criteria. These criteria include (1) displaying attenuated psychotic symptoms over the last year, (2) experiencing brief and limited intermittent psychotic symptoms, or (3) having a genetic vulnerability, which includes meeting the criteria for schizotypal personality disorder or having a first-degree relative diagnosed with a psychotic disorder [35]. While this third criterion is especially similar to schizotypy, there are important differences between these constructs. First, schizotypy is organized into positive, negative, and disorganized subtypes, whereas ultra-high-risk definitions tend to focus most on positive symptoms. Second, ultra-high-risk individuals are often required to have experienced clinical symptoms in the past year. In contrast, schizotypy can be identified psychometrically in individuals who are not displaying clinically meaningful symptoms. These two differences allow for a broader and earlier target for early intervention in schizotypy. This is especially relevant for individuals who may display more negative symptoms. As noted previously, negative schizotypy may be under-detected in research situations, which suggests that there are missed opportunities for early intervention among individuals with negative schizotypy as well.

The treatment of manifestations of schizotypy may represent an important component of early intervention for recovery from schizophrenia spectrum disorders. Traditional treatments often focus on pharmacological intervention to reduce symptomology [36,37] and skill-building to improve functional outcomes [38] in clinical schizotypy populations. Although one study found that an integrated treatment approach postponed the onset of psychosis in individuals with schizotypy [38], there is limited information available to use when making treatment decisions for individuals experiencing schizotypy or clinical manifestations of schizotypy, like schizotypal personality disorder [39]. We suggest that an existing framework used in the treatment of schizophrenia spectrum disorders, namely the integrated model of metacognition, can be applied to schizotypy populations as a targeted early intervention that promotes recovery.

## 3. Metacognition

At its core, metacognition refers to the ability to think about one’s own thinking and respond accordingly [40]. The integrated model of metacognition has expanded upon this definition to include the ability to understand and form increasingly complex and integrated ideas of oneself, others, and one’s community [41]. This model of metacognition can be understood as a spectrum or “umbrella” [42] of activities, ranging from discrete cognitive activities (e.g., recognizing a thought or emotion) to more complex functions, including social cognition and neurocognition, which may support more cohesive understandings of the self and others. The integrated model of metacognition, as proposed by Lysaker and colleagues, includes four domains of metacognition, including self-reflectivity, understanding the mind of the other, decentration, and mastery, each of which can be measured using the Metacognition Assessment Scale—Abbreviated (MAS-A) [43]—which was developed from the original Metacognition Assessment Scale (MAS) [44]. Although the MAS-A is often used to assess and measure domains of metacognition, metacognition can be measured using a wide range of psychometric tools [45].

The four domains of the integrated model of metacognition are thought to be distinct constructs yet still interact with one another in ways that promote reflection. Self-reflectivity is defined as the ability to understand the self in increasingly complex and integrated ways. In practice, self-reflectivity may range from being able to identify a discrete thought or emotion to being able to form a multifaceted and complex life narrative. Comparatively, understanding the mind of the other is understood as the capacity to understand specific others in increasingly complex and integrated ways. Decentration is defined as the ability to notice and understand that others in the world have complex lives and experiences that are unrelated to oneself. At lower levels of decentration, individuals may have the sense that they are at the center of others’ thoughts or display increased self-referential thinking. Finally, mastery is defined as the ability to use metacognitive knowledge about the self, others, and the world to respond to and manage psychological distress [41,46]. This may range from responding to psychological obstacles by largely evading them to employing a deeper understanding of one’s own mind, the minds of others, and the overall human experience to respond to challenges.

The integrated model of metacognition was initially developed to understand fragmentation in psychotic disorders and is most often applied to individuals diagnosed with schizophrenia spectrum disorders. However, there is existing evidence that metacognition is a transdiagnostic mechanism in various mental illnesses, including trauma-related disorders, personality disorders, substance use disorders, bipolar disorders, and depressive disorders. The self-regulatory executive function (S-REF) model provides one transdiagnostic framework that explains the role of metacognition in a range of psychological disorders [47]. This model identifies a universal style of negative processing that is linked to dysfunctional metacognition, thus providing additional support for the idea that deficits in metacognition may be a common feature across various mental illnesses.

Although metacognitive deficits have been observed in various mental illnesses, relatively greater deficits in metacognition have been observed in individuals with schizophrenia spectrum disorders in comparison to healthy controls [48], individuals facing medical adversity [49], and individuals with other psychiatric diagnoses, including bipolar disorder, depression, anxiety, posttraumatic stress disorder, substance use, and borderline personality disorder [50]. Metacognitive deficits have also been noted in individuals experiencing first-episode psychosis [51]. Importantly, impairment in metacognitive functioning has been observed internationally and across varying cultures in individuals diagnosed with schizophrenia spectrum disorders [50]. As noted above, there is evidence that impaired metacognition is present in personality disorders at varying levels. For example, metacognitive profiles of individuals diagnosed with borderline personality disorder show deficits in mastery and decentration [52,53]. A similar concept, mentalization, is also often studied in personality disorders, with most research focused on borderline personality disorder [54]. However, these constructs are less often studied in personality disorders more closely related to clinical expressions of schizotypy, such as schizotypal and schizoid personality disorders.

Metacognition is related to functional outcomes in numerous domains in schizophrenia spectrum disorders, including both objective and subjective measures of functioning. For example, metacognitive deficits have been linked with various neurobiological factors, cognitive disorganization, and psychological functioning [50]. In terms of psychological factors, deficits in metacognition have been connected to poor insight [55], social dysfunction [56], and poorer quality of life [43]. Although deficits in metacognitive functioning have been connected to positive, negative, and disorganized symptoms, research connecting metacognition to negative symptoms is especially robust. For example, research across international settings demonstrates that lower levels of metacognition are not only associated with more severe negative symptoms [43,51,57] but also predict later negative symptom severity in prospective studies [58,59].

Although deficits in metacognition have been linked to various measures of functioning, these impairments may be expressed in diverse ways. For example, the inability to understand others in complex and integrated ways may result in social alienation and dysfunction, whereas difficulty understanding one’s own thoughts, desires, and emotions may result in a lack of emotional experience, expression, and disturbances in first-person experience. Lower levels of mastery, or the inability to use metacognitive knowledge to respond to psychological problems, may result in a sense of hopelessness or helplessness. Broadly, with deficits in metacognition, thoughts and life narratives may be fragmented with the lack of a central theme or organizing factor, resulting in an understanding of the self and a world that is barren and relatively empty [50,60]. In contrast, improvements in metacognition have been connected to various improvements in both objective and subjective outcomes related to recovery in schizophrenia spectrum disorders [50,61]. For example, metacognitive capacity has been theorized to contribute to meaning-making, agency, and self-direction, thus promoting important aspects of subjective recovery [61,62]. Stated differently, intact metacognitive functioning results in individuals having a more integrated sense of their purposes and possibilities, as well as their position in the world, which contributes to a greater sense of meaning, agency, and quality of life.

### Metacognition in Schizotypy

Metacognition has also been proposed as an important transdiagnostic construct across a range of both clinical and non-clinical manifestations of schizotypy [25]. For example, it has been demonstrated that individuals with relatively high levels of schizotypy display metacognitive deficits in self-reflectivity and understanding the mind of the other, indicating that they may struggle to understand the self and others in cohesive and integrative ways [63]. Dysfunctional metacognitive beliefs have also been linked to higher levels of self-reported schizotypy, such that low-cognitive confidence and positive beliefs about worry predict self-reported schizotypy [64]. In this study, the high self-reported schizotypy group also displayed higher scores on all subscales of the Metacognitions Questionnaire-30, including cognitive self-consciousness, negative beliefs about uncontrollability and danger, and the need to control thoughts in addition to the two aforementioned subscales. Another study found evidence of a relationship between negative schizotypy, decentration, and an awareness of others [65]. Furthermore, one recent systematic review and empirical study concluded that higher rates of schizotypy were associated with various factors related to metacognitive deficits, including a maladaptive and overly self-referential metacognitive style and reduced introspective insight [66]. In the experimental part of this study, individuals with higher levels of schizotypy showed lower insight and maladaptive self-regulatory processes, which could be connected to the transdiagnostic S-REF model of metacognition discussed above.

Based on the findings reviewed above, we conclude that not only are metacognitive deficits present in schizotypy, but metacognition is a viable treatment target for individuals with schizotypy and related clinical disorders. We suggest that much of the research focused on metacognition in schizophrenia spectrum disorders can be applied to schizotypy. Although more research is needed, decrements in metacognition may also contribute to the progression from schizotypy to more severe manifestations of illness, including schizophrenia, while the amelioration of these deficits may enhance aspects of recovery, including social functioning, meaning-making, and the ability to form an integrated sense of self, others, and the wider world. For example, one recent study found that higher levels of schizotypy were significantly associated with the transition to a psychotic disorder from a clinical high-risk state, with negative schizotypy being the strongest predictor [67]. Another study found that negative schizotypy predicted increased aberrant salience in men but not women [68]. Taken together, the connections between negative schizotypy and the transition to a psychotic disorder and negative schizotypy and metacognitive deficits may elucidate the possibility that metacognitive interventions delay or prevent the transition to a schizophrenia spectrum or other psychotic disorder.

## 4. Early Intervention Treatment Implications

Although some interventions for schizotypy and related disorders may neglect important aspects of personal recovery, we propose that metacognitive-based treatments are uniquely able to place personal recovery at the center of treatment due to their focus on promoting an individual’s understanding of themselves, others, and their community. In addition, these treatments are likely to encourage individuals to make sense of their challenges and decide how they wish to move forward in their lives. Stated differently, approaches that utilize the integrated model of metacognition may encourage the development of a sense of purpose, possibility, and position in the world that further promotes subjective recovery [69,70]. As our understanding of the integrated model of metacognition has advanced, various recovery-oriented therapeutic approaches have emerged, including the Evolutionary Systems Therapy for Schizotypy (ESTS) and Metacognitive Reflection and Insight Therapy (MERIT). Of note, other treatments that are conceptually similar to metacognition-based treatment have been explored in clinically high-risk populations, including mentalization-based treatment [71].

One treatment that focuses on metacognition and was developed specifically for schizotypy is Evolutionary Systems Therapy for Schizotypy (ESTS) [72]. This approach combines evolutionary, metacognitive, and compassion-focused approaches to address schizotypy and schizotypal personality disorder. Individuals who engage in this metacognitive-oriented approach achieve improved functioning and reductions in symptoms and schizotypal features [73]. In a randomized controlled trial, individuals engaged in ESTS showed a larger decrease in symptomology and a greater increase in metacognition than an active control group [72]. ESTS has also shown promise in treating individuals diagnosed with other personality disorders related to schizotypy, including schizoid personality disorder [74].

Metacognitive Reflection and Insight Therapy (MERIT), another metacognitive treatment, is thought to promote recovery through a framework of preconditions and clinical elements, which include directly targeting metacognitive domains [46]. MERIT has a robust and expanding research base, which includes data from numerous randomized controlled trials, qualitative studies, case studies, and session-by-session evaluations that establish MERIT’s acceptability, feasibility, and effectiveness in treating schizophrenia spectrum disorders [75].

More relevant to the topic of early intervention, one clinical trial demonstrated that individuals diagnosed with early psychosis showed improved insight after receiving MERIT [76]. Case studies of MERIT have revealed improvements in a first-episode population in negative symptoms [77,78], positive symptoms [79,80], and depression [81]. One case study approached treatment using the idea that metacognitive deficits leave individuals without complex ideas of the self and others, leading to less goal-directed activities and a decreased ability to construct meaning in their lives, thus resulting in increased negative symptoms. This case study found improved metacognition and diminished negative symptoms over the course of 40 weeks of MERIT [77]. Although these studies focused on individuals already displaying symptoms of schizophrenia spectrum disorders, they support the idea that MERIT can be used in later stages of early intervention and may have unique promise for addressing early negative symptomology.

MERIT has also been successfully used as a treatment for individuals diagnosed with personality disorders, indicating its applicability to those with maladaptive personality organizations, such as schizotypy. These studies further demonstrate the potential for using MERIT for early intervention in schizotypy-related disorders. Relatedly, Dmitryeva et al. [82] reported on an individual diagnosed with schizotypal personality disorder who completed three months of MERIT and described decreased symptoms and distress and improved metacognitive functioning. One clinical case example reviewed the use of MERIT on an individual with elevated levels of schizotypy and noted improvements in symptomology, functioning, and metacognitive capacity [83]. In this case’s example, the importance of making meaning from one’s personal experiences was a central part of therapy, which further promoted recovery, as this individual found a personally acceptable sense of his challenges and decided how to respond to them to move forward in his life.

## 5. Discussion

Individuals with high levels of schizotypy represent a population of individuals who would likely benefit from early psychosocial intervention. These individuals may display a wider range of clinical indicators for the development of a schizophrenia spectrum disorder, including changes in positive, negative, and disorganized domains of functioning, and may be identified earlier in the progression to clinical symptomology. Unlike ultra-high-risk individuals whose identification largely hinges on the potential progression to an initial episode of psychosis, those with clinical or high-level manifestations of schizotypy can be identified and provided with interventions based on the observable characteristics and symptoms independent of psychotic episodes. Those with relatively high levels of negative schizotypy may be especially important to consider within this framework. Given that negative schizotypy is associated with increased polygenic risk scores for schizophrenia, an increased risk of transition to a psychotic disorder, and greater deficits in metacognition, it appears critical to consider the importance of early metacognitive intervention in this population.

Metacognitive deficits in individuals with high levels of schizotypy may manifest as having difficulty in understanding and reflecting upon their own thoughts and feelings, as well as challenges in attributing mental states to others and difficulty managing psychosocial problems. Such metacognitive deficits can make it challenging for individuals with schizotypy to recognize and regulate their own experiences, emotions, and behaviors, likely impacting social functioning and creating difficulty in forming and maintaining meaningful relationships. Metacognitive deficits are prevalent in personality disorders and dysfunctional personality organizations, including schizotypy. Greater metacognitive deficits, in turn, increase psychopathological risk and may increase the risk of a transition to other disorders, including those on the schizophrenia spectrum. Based on these findings, an approach focused on metacognition may represent a protective factor against worsening symptoms and diagnostic migration.

Interventions that aim to enhance metacognitive capacity, such as MERIT or ESTS, have been shown to increase psychological insight, improve psychosocial functioning, and decrease schizotypy-related symptomology. These changes may then reduce the likelihood of progressing to more severe schizophrenia spectrum disorders, especially in cases of more severe negative schizotypy, which is often neglected in ultra-high-risk studies. By emphasizing meaning-making and an increased understanding of the self, others, and psychosocial challenges, metacognitive-based psychotherapies may uniquely promote subjective recovery in schizotypy. Of note, although we focused on two metacognitive-based treatments (MERIT and ESTS), other metacognitive psychotherapies not covered here have been developed and may be similarly beneficial. In addition, treatments that utilize similar constructs (e.g., mentalization) may also be relevant to this work. Furthermore, focusing on metacognition as a determinant of recovery that can be targeted in psychosocial interventions may bypass some of the common controversies of treating ultra-high-risk individuals, including concerns about possible long-term side effects of medications and the potential for stigmatization and coercion [84]. This approach also aligns with the early intervention clinical staging model proposed by McGorry and colleagues [19], such that more benign treatments are being used in early interventions for high-risk individuals.

## 6. Conclusions

Individuals identified as experiencing high levels of schizotypy are at increased risk of developing a schizophrenia spectrum disorder, ranging from schizotypal personality disorder to schizophrenia. The early identification of these individuals, coupled with early intervention, may pave the way for improved recovery outcomes. Metacognitive deficits, including difficulty understanding and forming increasingly complex ideas about the self, others, and the wider community, are well-established in schizophrenia spectrum disorders and have also been observed in individuals with high levels of schizotypy. Thus, metacognition may represent one transdiagnostic construct that can be directly targeted with psychotherapy, including MERIT and ESTS, to enhance recovery and protect against diagnostic migration.

## 7. Future Directions

This paper is largely theoretical; therefore, empirical research to test these claims is still needed. For example, more research is needed to establish metacognitive-based deficits and explore the impact of metacognitive interventions in schizotypy populations. This may include randomized controlled trials and studies that focus on what elements of metacognitive interventions are most effective in promoting objective and subjective domains of recovery and what dimensions of schizotypy may be most impacted by metacognitive intervention. In addition, more research is needed that addresses the idea that metacognitively based interventions may prevent or delay the transition to more severe schizophrenia spectrum disorders in schizotypy populations.

## Data Availability

Not applicable.

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
