# Peer review of "Metacognition as a Transdiagnostic Determinant of Recovery in Schizotypy and Schizophrenia Spectrum Disorders"

_behavsci, 2024, doi:10.3390/bs14040336_

Round 1

Reviewer 1 Report

Comments and Suggestions for Authors

Dear authors,

Thank you very much for your submission to Behavioral Sciences. Given the theoretical nature of your review - that is, the lack of 'scientific methodology'-, I have no comments to be addressed.

Yours sincerely,

Reviewer X

Comments on the Quality of English Language

As noted above, I have recommended the authors to review the English (minor editing).

Author Response

Thank you for taking the time to review our manuscript. We have carefully proof-read the manuscript for editing and clarity. We appreciate your feedback and hope that our changes have improved the manuscript. 

Reviewer 2 Report

Comments and Suggestions for Authors

The article is interesting, especially for the focus on metacognition as a transdiagnostic construct and as an early and protective psychotherapeutic treatment.

Some important fixes and suggestions:

From line 102 to line 122, To clarify:

By definition, the term schizotypy refers to a set of dysfunctional traits and is closely associated with related personality disorder. You need to clarify better what you mean by "adaptive traits" and also the relationship between these traits and the disorder, in its dimensional evolution.

Furthermore, you should better specify the dimensionality and evolution in terms of risk. This is because the percentage of schizotypal patients who develop a schizophrenic disorder is small (DSM 5 and 5 TR among others)

Clarifying is important, also because in the continuation of the discussion (from line 125) you talk about sub-clinical traits with an impact on functioning, distinguishing them from "without impact" traits (happy schizotypes). The bibliographical references are there, but you must clarify in the body of the text.

From line 202, expand the description a little. Especially compared to the fact that metacognitive models are also widely used for understanding personality disorders, which by definition present impaired metacognitive characteristics, specifically associated with individual PDs

In the paragraph on metacognition in schizotypy, I would give more details on the metacognitive deficits which, evidence based, appear more specifically associated with schizotypal traits.

In general, argue more about the idea of metacognitive difficulty as a transdiagnostic construct

I believe that, in order to support the conclusions, it should be clarified better what the criteria are for distributing schizotypy traits across various risk levels, according to the authors. In the text there is a small part dedicated to this but it should be better discussed and supported, precisely to validate the clinical reasoning on early work and metacognition.

From 342: “Because metacognitive impairments have been uniquely linked to negative symptoms across the schizophrenia spectrum, including in schizotypy, targeting metacognition via early intervention in this population may help delay the progress of symptomology or transition to a psychotic disorder…”

Here too, to clarify: I would rather say: metacognitive deficits are highly prevalent in personality disorders and dysfunctional personality traits. A greater expressive intensity of these characteristics increases the psychopathological risk and increases the risk of transition to other disorders, such as those of the psychotic spectrum. A work focus centered on metacognition and specific for characteristics can therefore represent a protective factor from worsening and diagnostic migration.

Reviewer 3 Report

Comments and Suggestions for Authors

The article deals with the crucial issue of early intervention not only for early phase psychosis patients but also for individuals at high risk of developing the illness such as schizotypic individuals. The authors propose a fascinating hypothesis according to which metacognition-based intervention structured for schizotipy may reduce the risk of transition to more severe forms of schizophrenia spectrum disorders.

Research on metacognitive psychotherapies is on the rise, with a strong need for intervention in high-risk individuals. The hyphotesis sounds very intriguing from this perspective, and may provide a starting point for future research. The article is well written and articulated. Nevertheless it may improve in clarity with some minor revisions as follows:

Introduction

Lane 30 Please substitute has supported with “supports”

Lane 59 “after the onset”

Lane 65 “improving recovery”

Lane 83-85 “It is further suggested that benign interventions (e.g., fatty acids, psychotherapy) be used in high-risk populations while more risky and complex interventions (e.g., psychotropics) be used in later stages or when the aforementioned have failed”

Though formally correct, please rewrite the above sentence as it can imply that pharmacological treatments are necessarily harmful; at the same time a psychoterapeutic intevention may be not simple in high risk psychosis individuals.

Lane 88 substitute “for developing” with “of developing”

Lane 92-93 The sentence is not clear. Please reformulate

Metacognition

Authors focused their attention on the theoretical integrated model of metacogniton that might be linked to what they want to speculate on (schizotipy and metacognition)

Nevertheless, without deeply speculating about metacognition conceptualization, I think it would be appreciated and useful by the interested reader to cite (without necessarily speculate on) the “umbrella concept” expressed by Zita Fekete et al. 2022 and the S-Ref model of metacognition. In the same way, it should be mentioned that metacognition in schizophrenia and related disorders can be evaluated throug a wide range of psychometric tools as recently reviewed by Martiadis et al. 2023 (Frontiers in Psychiatry).

“Interventions that aim to enhance metacognitive capacity, such as MERIT or ESTS, may increase psychological insight, improve psychosocial functioning, decrease symptoms, and reduce the likelihood of progressing to more severe schizophrenia spectrum disorders, especially in cases of more severe negative schizotypy, which is often neglected in ultra-high risk studies”

This sentence might be reformulated as there is still no evidence that MERIT and ESTS can "reduce the likelihood of progressing to a more severe schizophrenia form of disorder". Authors should specifically address that the positive effects of these therapies (increased insight, improved functioning, etc.) which are already demonstrated, should help to reduce the likelihood of progressing (which is to be demonstrated).

Authors should note that, despite focusing on the two approaches mentioned, other metacognition-based psychotherapeutic approaches (with different theoretical underpinnings) have been developed recently, as Phillip (2018) describes; it can be useful for the reader interested in metacognitive psychotherapeutic treatment, and merely mentioning these approaches does not go beyond the scope of this paper.

Lane 364 “metacognitive-based”

Lane 363-366 Please rewrite to improve clarity.
